# Inversion of Coniferous Forest Stock Volume Based on Backscatter and InSAR Coherence Factors of Sentinel-1 Hyper-Temporal Images and Spectral Variables of Landsat 8 OLI

**Xinyu Li** [1,2,3,4,†], **Zilin Ye** [1,3,4,†], **Jiangping Long** [1,3,4], **Huanna Zheng** [1,3,4] and **Hui Lin** [1,3,4,*]

1 Research Center of Forestry Remote Sensing & Information Engineering, Central South University of Forestry and Technology, Changsha 410004, China; lxy365@csuft.edu.cn (X.L.); 20191100019@csuft.edu.cn (Z.Y.); t20080976@csuft.edu.cn (J.L.); 20211100037@csuft.edu.cn (H.Z.)
2 School of Computer Science, Hunan First Normal University, Changsha 410205, China
3 Key Laboratory of Forestry Remote Sensing Based Big Data & Ecological Security for Hunan Province, Changsha 410004, China
4 Key Laboratory of State Forestry Administration on Forest Resources Management and Monitoring in Southern Area, Changsha 410004, China
* Correspondence: t19911090@csuft.edu.cn; Tel.: +86-0731-8562-3848
† These authors contributed equally to this work.

**Abstract:** Forest stock volume (FSV) is a basic data source for estimating forest carbon sink. It is also a crucial parameter that reflects the quality of forest resources and forest management level. The use of remote sensing data combined with a support vector regression (SVR) algorithm has been widely used in FSV estimation. However, due to the complexity and spatial heterogeneity of the forest biological community, in the FSV high-value area with dense vegetation, the optical re-mote sensing variables tend to be saturated, and the sensitivity of synthetic aperture radar (SAR) backscattering features to the FSV is significantly reduced. These factors seriously affect the ac-curacy of the FSV estimation. In this study, Landsat 8 (L8) Operational Land Imager multispectral images and C-band Sentinel-1 (S1) hyper-temporal SAR images were used to extract three re-mote sensing feature datasets: spectral variables (L8), backscattering coefficients (S1), and inter-ferometric SAR factors (S1-InSAR). We proposed a feature selection method based on SVR (FS-SVR) and compared the FSV estimation performance of FS-SVR and stepwise regression analysis (SRA) on the aforementioned three remote sensing feature datasets. Finally, an estima-tion model of coniferous FSV was constructed using the SVR algorithm in Wangyedian Forest Farm, Inner Mongolia, China, and the spatial distribution map of coniferous FSV was predicted. The experimental results show the following: (1) The coherence amplitude and DSM data ob-tained based on S1 images contain information relat-ed to forest canopy height, and the hy-per-temporal S1 image data significantly enrich the diversity of S1-InSAR feature factors. There-fore, the S1-InSAR dataset has a better FSV response than remote sensing factors such as the S1 backscattering coefficient and L8 vegetation index, and the corresponding root mean square er-ror (RMSE) and relative RMSE (rRMSE) values reached 47.6 m$^3$/ha and 20.9%, respectively. (2) The integrated dataset can provide full play to the synergy of the L8, S1, and S1-InSAR remote sensing data. Its RMSE and rRMSE values are 44.3 m$^3$/ha and 19.4% respectively. (3) The proposed FS-SVR method can better select remote sensing variables suitable for FSV estimation than SRA. The average value of the rRMSE (23.17%) based on the three datasets was 13.8% lower than that of the SRA method (26.87%). This study provides new insights into forest FSV retrieval based on active and passive multisource remote sensing joint data.

**Keywords:** forestry remote sensing; forest stock volume; feature variable selection; synthetic aperture radar; InSAR coherence





## 1. Introduction

The carbon stored through forest ecosystems accounts for about 2/3 of the total carbon pool of the whole terrestrial ecosystems [1,2]. Forest stock volume (FSV) is a basic data source for estimating forest biomass and the carbon sink. It is also a crucial parameter that reflects the quality of forest resources and forest management level [3–5]. The dynamic estimation of the spatial distribution of FSV is not only the basis of scientific and accurate forest management, but also the prerequisite for maximizing the function and carbon se-questration potential of forest ecosystems [6–8].

Although many large-scale global products can easily obtain FSV information, the use of specific areas is easily limited by time-frequency, spatial resolution, or other unknown local errors [9–15]. Due to the strong spatial heterogeneity of forest ecosystems, the response relationship between remote sensing factors and FSV is usually complex [16,17]. Many problems remain in the research and application of FSV estimation based on remote sensing technology. Optical remote sensing is a passive remote sensing technology. In the densely vegetated high-value area of the FSV, optical remote sensing technology cannot obtain the spectral signal of the inner section of the forest and the vertical direction of the canopy; therefore, problems such as poor spectral sensitivity and low light saturation point are observed [18–23]. LiDAR is an emerging active remote sensing technology [24–30]. However, few spaceborne data are available, the cost of airborne data acquisition is high, which limits its widespread application in the field of FSV remote sensing estimation [20]. Microwave radar is an advanced active remote sensing technology not easily affected by climatic factors, such as clouds, fog, and solar radiation, and can realize all-weather earth observations [31–33]. Depending on the wavelength and frequency, microwave radar signals can penetrate the forest canopy to different degrees and obtain comprehensive in-formation on the forest structure at different levels and orientations [34–39]. The microwave radar remote sensing technologies commonly used in FSV estimation include synthetic aperture radar (SAR), interferometric radar (InSAR), and polarization interferomet-ric radar (PolInSAR) [40,41].

The microwave radar backscattering coefficient and its textural features have been widely used in research on FSV and forest aboveground biomass (AGB) [39,42]. Erkki et al. [43] used Sentinel-1 C-band SAR to conduct a forest snow damage mapping study in northern and southern Finland, with an overall accuracy of 90%. They also estimated the FSV in damaged forest areas, and the results suggested that multitemporal Sentinel-1 data have good potential for estimating the overall FSV. In research on FSV estimation based on C-band-and L-band SAR images, the findings of Tanase et al. [44] demonstrated that the FSV estimation performance of C-band and L-band SAR data is almost the same, and the synergy between the two data is limited. Purohit et al. [40] used Landsat 8 OLI and Senti-nel-1A images to accurately predict the spatial distribution of AGB of different forest types in the foothills of the Indian Himalayas, indicating that the coordination of optical remote sensing variables and radar backscatter data can effectively improve the accuracy of forest AGB estimation. Using InSAR technology to perform radar signal interferometric processing on two SAR complex images can generate an interferometric phase, interferometric coherence coefficient, digital surface model (DSM), and other feature factors that contain information on the horizontal and vertical structures of the forest, which are usually very beneficial for FSV estimation [42]. Borlafmena et al. [45] assessed the utility of Sentinel-1 coherence time series for temperate and tropical forest mapping. They found that for forest classification on rough terrain, the Sentinel-1 coherence amplitude can significantly reduce the error of forest missions. In addition, Sentinel-1 time series data based on InSAR technology have shown excellent performance in the research of land subsidence, surface deformation monitoring, and post-disaster assessment [41,46–49]. Sentinel-1 imagery has the advantages of global coverage, free access, and high spatial and temporal resolution [42,45]; however, the full potential of Sentinel-1 C-band interferometry SAR (S1-InSAR) coherent data for estimating boreal FSV has rarely been studied. In addition, the Sentinel-1 backscattering coefficient,

InSAR coherence coefficient, and Landsat 8 OLI spectral feature variables were not used together to predict FSV.

The selection of remote sensing feature variables is a key factor in determining the accuracy of the FSV estimation model [20,50–53]. The random forest (RF) method can evaluate and rank the importance of remote sensing features by using the out-of-bag esti-mation error before and after adding noise to the feature variables; however, it does not consider the combination effect relationship between the feature variables [10,12,50]. Stepwise regression analysis (SRA) can dynamically eliminate redundant features through a variance homogeneity test; however, it can only select variables based on the linear relationship between remote sensing variables and FSV [10]. The support vector regression (SVR) algorithm is a classical small-sample learning method with a profound theoretical foundation. It has a satisfactory generalization performance, is resilient to overfit, and can achieve good estimation performance in the case of very few sample data. Therefore, this method has been widely used for remote sensing monitoring and modeling of forest resources [20,52,54]. However, few studies have used the SVR algorithm combined with the combination effect of remote sensing feature variables to select remote sensing variables.

Therefore, this study will build a coniferous plantation FSV remote sensing inversion experimental area in northern China. Based on sentinel-1 SAR backscattering coefficient, hyper temporal InSAR coherence factors and Landsat 8 spectral variables, SVR algorithm combined with the combination effect of remote sensing characteristic variables will be used to select remote sensing variables and build FSV estimation model, so as to effective-ly improve the estimation accuracy of coniferous forest volume through the combination of active and passive remote sensing data.

## 2. Study Area and Data

### 2.1. Study Area

The study area, the Wangyedian Forest Farm, with a forest area of 23,118 ha, is located in Harqin, Inner Mongolia Autonomous Region, Northeast China (118°09′ to 118°30′E, 41°21′ to 41°39′N) (Figure 1). This choice was influenced by the temperate monsoon climate as well as the annual precipitation, temperature, and the frost-free period of the study area, which are, approximately, 400 mm, 4.2 °C, and 117 d, respectively. There are many rolling mountains, with an altitude distribution of 800–1890 m, on the forest farm. The forest farm is rich in forest resources and beautiful scenery, with a total FSV of 1.5 million m$^3$, of which the FSV of the plantation is approximately 0.8 million m$^3$. It has a forest area of 350,000 mu, including 176,000 mu of planted forest, and the main tree species are larch (Larix gmelinii Kuzen.) and Chinese pine (Pinus tabuliformis Carrière). The natural forest area is 174,000 mu, and the main tree species are white birch (Betula platyphylla Suk.), aspen (Populus davidiana Dode) and oak (Xylosma racemosum Miq.) [12].

### 2.2. Sample Plot Design and FSV Data Collection

The setting of the FSV sample plot was selected according to many factors, including altitude, slope direction, slope, and stand age structure. Each sample plot had a size of 25 × 25 m, was required to contain only one main forest type, and was far from the stand boundary. For fulfilling the needs of high-precision positioning and measurement, the investigators used the RTK to collect the coordinate position and terrain information in the sample plot and measured the height of trees in the sample plot with a laser altimeter; the DBH of all living trees with a DBH greater than or equal to 5 cm in the sample plot was measured with a special DBH ruler (1.3 m from the ground). The collected data included information on tree species, DBH, tree height, crown width, and the height of trees under branches, as well as the geographical environment data of the sample plot, such as the coordinates of the central point of the sample plot, altitude, slope, and slope direction.

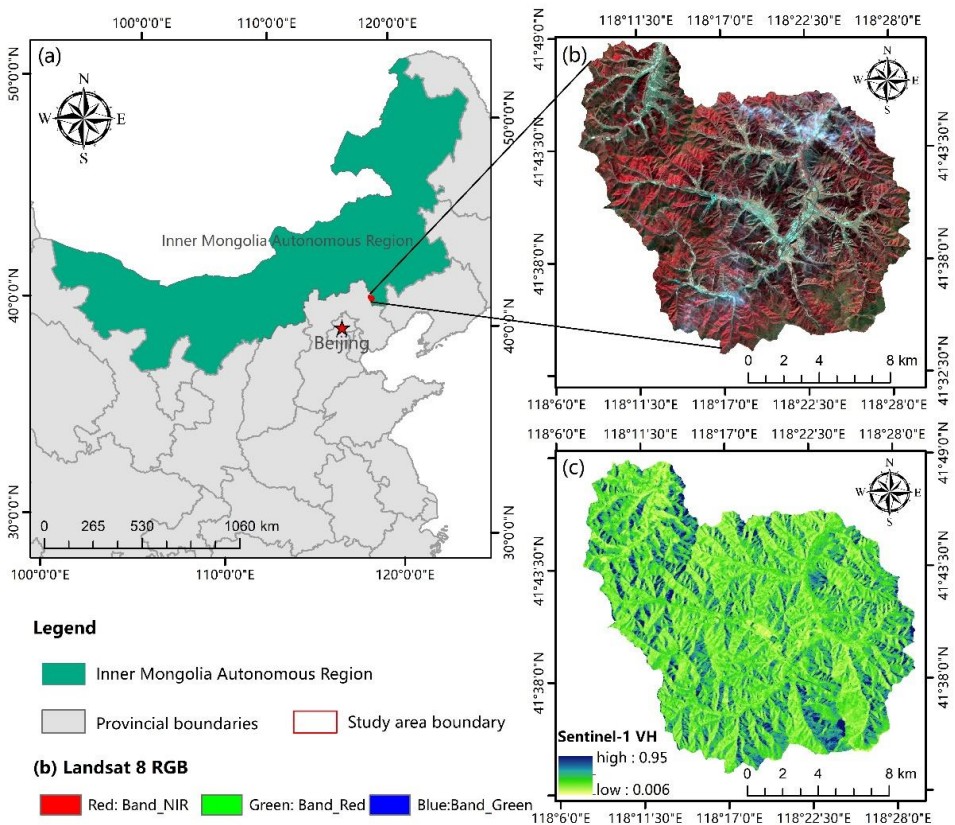

**Figure 1.** (**a**) Study area location; (**b**) Landsat 8 image of the study area; (**c**) Sentinel-1 C-band SAR image of the study area.

The fieldwork inventory was conducted in the autumn of 2017 and 2019, and there were 54 larch and 76 Chinese pine sample plots (Figure 2). Using the FSV calculation formula for larches and Chinese pines provided by the forest farm (Table 1), the stock volume of each tree in the sample plot was calculated. According to the growth curves of tree height and the DBH of larches and Chinese pines at different age levels, the corresponding annual relative growth was calculated to convert the FSV in 2017 into the FSV in 2019. The final FSV data from the 130 sample plots were summarized and converted into hectares ($m^3$/ha) (Table 2).

**Table 1.** Coniferous forest stock volume calculation formulas in the study area.

| Tree Species | FSV Calculation Formula | Remarks |
|---|---|---|
| Larch | $FSV = -0.001498 + 0.00007 \times D^2 + 0.000901 \times H + 0.000032 \times H \times D^2$ | |
| Chinese pine | $FSV = 0.013464 - 0.001967 \times D + 0.000089 \times D^2 + 0.000628 \times D \times H + 0.000032 \times H \times D^2 - 0.003173 \times H$ | D: DBH H: Tree height |

**Table 2.** Statistical summary of forest stock volume ($m^3$/ha) at field plots.

| Tree Species | Numbers of Plots | Minimum | Maximum | Mean | Standard Deviation | Coefficient of Variation (%) |
|---|---|---|---|---|---|---|
| Chinese pine | 76 | 105 | 519 | 230.8 | 83.2 | 36.1 |
| Larch | 54 | 76 | 360 | 223.4 | 63.5 | 28.4 |
| All | 130 | 76 | 519 | 227.7 | 75.5 | 33.2 |

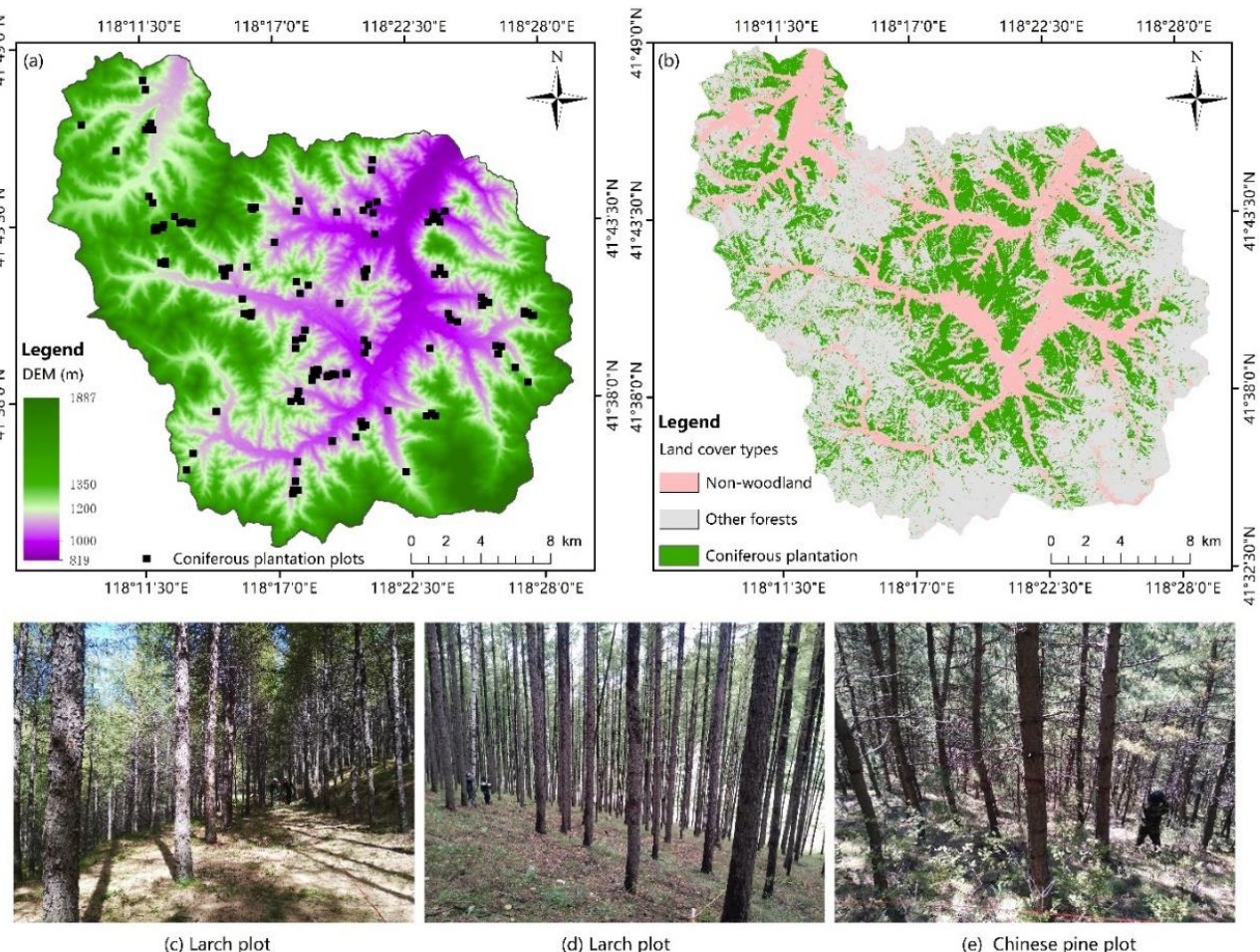

**Figure 2.** (**a**) Plot distribution map and digital elevation model (DEM) of the study area; (**b**) distribution of coniferous forest, other forest types, and non-forest land in the study area; (**c–e**) images of larches and Chinese pines at the field survey sites, respectively.

### 2.3. Optical and SAR Remote Sensing Data Preprocessing

One Landsat 8 (L8) and thirty-nine Sentinel-1 (S1) hyper-temporal images (Table A1) were obtained from the United States Geological Survey (https://earthexplorer.usgs.gov/) (accessed on 9 May 2021) and the European Space Agency Copernicus data center (https://scihub.copernicus.eu/) (accessed on 11 July 2021), respectively. Digital elevation model (DEM) data covering the study area were downloaded from the geospatial data cloud (http://www.gscloud.cn/) (accessed on 8 May 2019). They were generated using ASTER GDEM data processing, with a spatial resolution of 30 m, a projection type of UTM/WGS84, and a data type of IMG. We also obtained the forest resource distribution map, administrative boundary vector map, and other relevant auxiliary information pro-vided by the forest farm.

The acquired Landsat 8 Operational Land Imager (OLI) images (level-1T) underwent systematic radiometric and geometric corrections. The spectral information of the forest vegetation can be obtained from OLI images. However, to quantitatively retrieve forest FSV data using image spectral variables, Landsat 8 level-1T products usually need pre-processing, such as radiation calibration and atmospheric correction [10]. Sentinel-1 was the first C-band dual-polarization SAR satellite developed by the Copernicus program of the European Space Agency. Sentinel-1 A and Sentinel-1 B were successfully launched in April 2014 and 2016, respectively. Sentinel-1 images have the advantages of ultrahigh radiation resolution (1 dB/3σ), global coverage, free download, and high spatial resolution. Therefore, Sentinel-1 is widely used in all-weather and all-time radar imaging Earth observation missions [42,47]. The flight orbit space of the Sentinel-1 interferometric wide

amplitude mode (IW) is a pipe with a radius of approximately 50 m, and it generates three sub-band (IW1, IW2, IW3) data through progressive terrain scanning to ensure that the interferometric image data have good coherence; thus, it can be used to conduct effective radar interferometric analysis [42,55]. This study obtained 39 Sentinel-1 level-1 IW PolSAR images covering the Wangyedian Forest Farm from March to December 2019 (Table A1), which were single-view complex (SLC) products based on the raw data of the radar system after fast-focusing processing. Sentinel-1 SLC images require a series of processes, such as orbit correction, radiometric calibration, multi-looking, polarization filtering, terrain correction, and geocoding, to obtain a SAR backscattering coefficient. In addition, the 39 hyper-temporal SLC images can be used to form multiple image pairs, and canopy elevation data and coherent information can be obtained through InSAR technology.

## 3. Methods

### 3.1. Optical Remote Sensing Feature Variable Extraction

In research on forest FSV estimation based on optical images, the usually extracted optical remote sensing factors mainly include spectral band reflectance, a vegetation index, and texture feature factors extracted by image spatial texture analysis [10,21]. The vegetation index can realize the simple and effective measurement of surface vegetation growth trends, health status, and other information [12,20]. Through texture analysis, the periodic changes of the arrangement and combination attributes of texture primitives at the gray level of the image can be reflected macroscopically [56–59]. A gray level concurrency matrix (GLCM) is a typical statistical method for texture information analysis [52].

In total, 276 optical remote sensing feature variables were extracted from the L8 OLI images for forest FSV estimation (Table 3); among them, there were 7 multispectral bands, 45 vegetation index feature variables, and 224 texture feature factors. GLCM was used to obtain image texture information, such as the mean, variance, and homogeneity [13]. As shown in Table 3, based on the GLCM method, texture feature data were extracted from seven multispectral bands of the L8 OLI image, with a sliding window of $3 \times 3$, $5 \times 5$, $7 \times 7$, $9 \times 9$, and step size of [1,1].

**Table 3.** Optical remote sensing feature variables extracted from L8 OLI image.

| Variable Type | Variable Name | Variable Description |
|---|---|---|
| Band reflectance | Band i, (i = 1, . . . , 7) | Band 1: Coastal, Band 2: Blue, Band 3: Green, Band 4: Red, Band 5: NIR, Band 6: SWIR 1, Band 7: SWIR 2 |
| Vegetation indices | NDVI | $(\text{Nir} - \text{Red})/(\text{Nir} + \text{Red})$ |
| | RVI_ij | Band i/Band j, $i \neq j$ |
| | DVI_ij | Band i/Band j, $i \neq j$ |
| | EVI | $2.5 \times (\text{NIR} - \text{Red})/(\text{NIR} + 6 \times \text{Red} - 7.5 \times \text{Blue} + 1)$ |
| | SAVI | $(\text{Nir} - \text{Red})(1 + 0.5)/(\text{Nir} + \text{Red} + 0.5)$ |
| Texture (GLCM) | Mean (M) | $\sum_{i,j=0}^{n-1} i P_{i,j}$ |
| | Variance (Var) | $\sum_{i,j=0}^{n-1} P_{i,j}\left(i, j - \mu_{i,j}\right)$ |
| | Contrast (Con) | $\sum_{i,j=0}^{n-1} P_{i,j}(i-j)^2$ |
| | Homogeneity (Hom) | $\sum_{i,j=0}^{n-1} P_{i,j}/\left(1 + (i-j)^2\right)$ |
| | Entropy (Ent) | $\sum_{i,j=0}^{n-1} P_{i,j}\left(-\ln P_{i,j}\right)$ |
| | Dissimilarity (Dis) | $\sum_{i,j=0}^{n-1} P_{i,j}|i-j|$ |
| | Second Moment (SM) | $\sum_{i,j=0}^{n-1} \left(P_{i,j}\right)^2$ |
| | Correlation (Cor) | $\sum_{i,j=0}^{n-1} P_{i,j}\left((i - \mu_i)\left(j - \mu_j\right)/\sqrt{\sigma_i^2 \sigma_j^2}\right)$ |

Note: $P_{i,j} = V_{i,j}/\sum_{i,j=0}^{n-1} V_{i,j}$, is the normalized value of element $V_{i,j}$ in row $i$ and column $j$ of the GLCM, and N is the number of rows (columns) in the matrix.

### 3.2. SAR Feature Variable Extraction

In the microwave radar remote sensing system, ground object types usually include discrete targets, distributed targets, and targets that combine discrete and distributed targets, whereas forests usually belong to the third type of targets. For non-discrete targets, such as forests and farmland, the radar image unit pixel usually contains many scatterers; therefore, the radar echo signal is formed by the coherent superposition of all scatterer signals [39]. Such coherently superimposed scattered signals usually indicate the absence of dominant scatterers; therefore, the concept of the backscattering coefficient needs to be introduced to describe them [41]. The backscattering coefficient is the reflectivity of the radar electromagnetic wave signal per unit of the backscattering cross-sectional area. It describes the interaction between the incident radar electromagnetic waves and ground objects by using statistical methods to measure the scattering ability of ground objects [46]. As shown in Equation (1), the backscattering coefficient Sigma$^0$ ($\sigma^0$) can be expressed as the average scattering cross-section corresponding to the unit-effective scattering unit area, which is a dimensionless quantity [39,42,44].

$$\sigma^0 = \sigma / Area_\sigma \tag{1}$$

where $\langle \sigma \rangle$ is the average scattering cross-section of the radar, and $Area_\sigma$ is the effective scattering area in the ground distance direction. In a specific application, the unit-effective scattering unit area can also be expressed as the effective scattering unit area $Area_\gamma$. perpendicular to the incident direction or the effective scattering unit area $Area_\beta$ in the oblique distance direction.

$$Area_\beta = Resolution_r \times Resolution_a = (c \times \tau/2) \times (D/2) \tag{2}$$

$$Area_\sigma = Resolution_r \times Resolution_a / \sin\theta = c \times \tau \times D / 4\sin\theta \tag{3}$$

$$Area_\gamma = Resolution_r \times Resolution_a / \tan\theta = c \times \tau \times D / 4\tan\theta \tag{4}$$

where $Resolution_r$ and $Resolution_a$ are the range and azimuth resolutions of the image, respectively, $c$ is the speed of light, $\tau$ is the pulse duration of the radar system, $D$ is the radar aperture size, and $\theta$ is the ground-incident angle corresponding to the image pixel. Therefore, the backscattering coefficient can be expressed in the form of Gamma$^0$ ($\gamma^0$) and Beta$^0$ ($\beta^0$). The radar feature variables extracted from Sentinel-1 SAR images in this paper are shown in Table 4.

$$\gamma^0 = \sigma / Area_\gamma \tag{5}$$

$$\beta^0 = \sigma / Area_\beta \tag{6}$$

**Table 4.** Radar feature variables extracted from Sentinel-1 SAR image.

| Variable Type | Variable Description |
|---|---|
| Backscattering coefficients | VV(Sigma$^0$), VH(Sigma$^0$), VV(Gamma$^0$), VH(Gamma$^0$), VV(Beta$^0$), VH(Beta$^0$) |
| Radar indices | $(VH - VV)/(VH + VV)$, $VV/VH$ |
| Texture features extracted from Backscattering coefficients | Mean (M), Variance (Var), Homogeneity (Hom), Contrast (Con), Dissimilarity (Dis), Entropy (Ent), Second moment (SM), Correlation (Cor) |

Note: VV and VH are the backscattering coefficients corresponding to the polarization modes of VV and VH, respectively; the sliding window scale for extracting texture features based on the backscattering coefficients is (3 × 3, 5 × 5, 7 × 7, 9 × 9, 11 × 11, 13 × 13, 15 × 15).

### 3.3. InSAR Feature Variable Extraction

InSAR is a space–Earth observation technology developed from traditional microwave radar remote sensing technology combined with radio astronomical interferometry [45,46]. Based on the microwave signals transmitted and received by the InSAR system from the

target area, SAR complex image pairs containing intensity and the phase information of ground objects in the same area can be generated. If there are coherence conditions between the complex image pairs, an interferogram can be generated using conjugate multiplication. The distance difference between the microwaves can be calculated according to the phase value of the interferogram. The distance information between the ground object target and the sensor can be obtained using parameters such as the flight altitude information of the satellite sensor, the frequency of the microwave radar, and the beam direction; consequently, elevation information corresponding to each pixel target in the SAR image can be measured accurately [48,49].

In this study, the VV and VH polarization data of Sentinel-1 SLC images were used for interference processing. As shown in Figure 3, in the process of interference processing using SNAP software, the downloaded DEM and the boundary shape file of the study area must be used to register and cut the Sentinel-1 image pair. The ratio of the distance view to the azimuth view was set to 5:1, and the mapping resolution was approximately 30 m. The Goldstein algorithm was selected as the filtering method of the interferogram, which not only improved the definition of the interference fringe but also effectively reduced the incoherent noise caused by errors. The phase unwrapping method uses the minimum cost flow method to mask pixels whose coherence is less than the threshold. A polynomial optimization method was used for orbit refining and phase offset correction. Based on the selected control points, re-flattening processing was conducted. Finally, geographic coding was conducted while referring to the DEM image coordinate system to obtain the DSM and the coherence coefficient map.

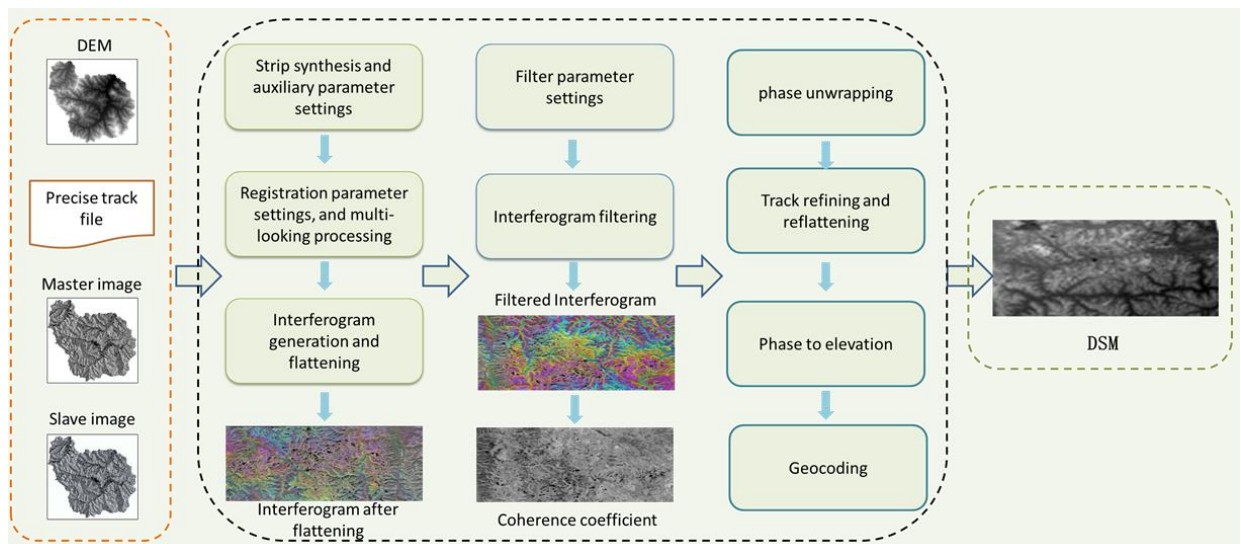

**Figure 3.** Flow chart of interference processing based on Sentinel-1 IW SLC images.

As shown in Table A2, 39 pairs of Sentinel-1 IW SLC images were sorted according to the imaging time interval, of which 29 pairs, 9 pairs, and 1 pair were separated by 12, 24, and 36 d, respectively. Because the spatial baseline length of some image pairs is too short (<15 m), the vertical distance is too small; the influence of the terrain is too large, which can easily lead to large interferometric errors, and interference processing cannot be performed. Some intermediate results obtained by the InSAR processing are shown in Figure 4.

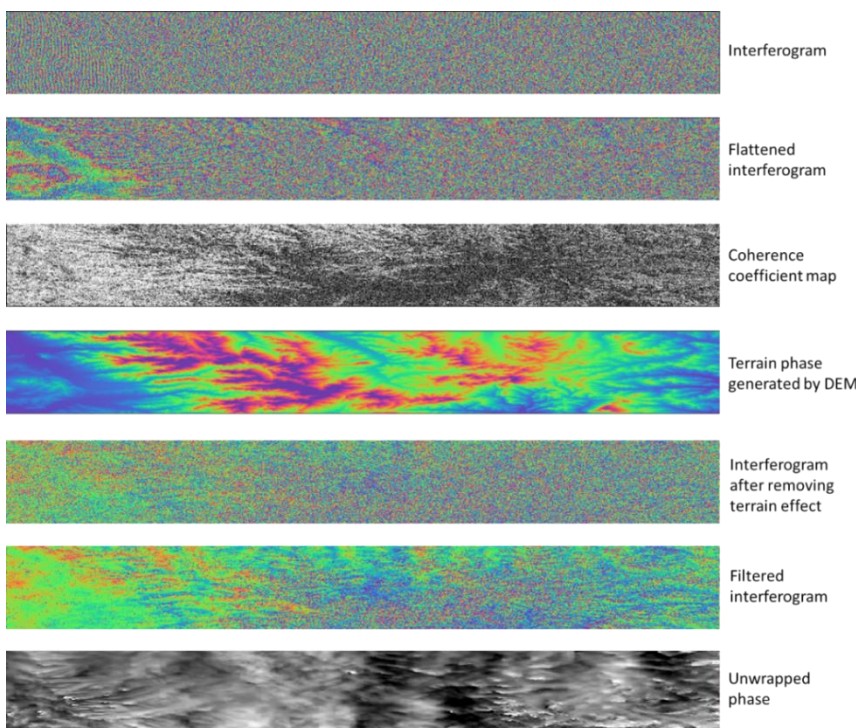

**Figure 4.** Some intermediate results obtained by radar interferometry based on Sentinel-1 images.

*3.4. Feature Variable Selection Method Based on SVR*

The SVR algorithm is a commonly used statistical learning method. For dataset D = {$(x_1,y_1),(x_2,y_2), \ldots ,(x_n,y_n)$}, with n training samples, $x_i$ is the independent variable with a p-dimensional feature space, and $y_i$ is the dependent variable. A regression function (Equation (7)) can be obtained using SVR learning such that $f(x)$ is as close to $y$ as possible, where $\omega$ and $b$ are the parameters to be solved. For all samples, the estimated loss was calculated if and only if the absolute value of the deviation between $f(x)$ and $y$ was greater than $\varepsilon$. At this point, it is equivalent to building an interval pipeline with a deviation range of $\varepsilon$, using $f(x)$ as the center [54,60]. Therefore, the optimization problem of SVR can be expressed using the following formula:

$$f(x) = \omega^T x + b \tag{7}$$

$$\min_{\omega,b} \frac{1}{2}\|\omega\|^2 + C \sum_{i=1}^{m} l_\varepsilon (f(x_i) - y_i) \tag{8}$$

where $C$ is the regularization constant and $l_\varepsilon$ is the loss function.

Based on the estimation error of the SVR model, this study explored the combinatorial optimization effect between feature variables and used the forward heuristic increment method to gradually select the appropriate feature variables from the set of candidate feature variables (FC).

First, the distance correlation measurement method was used to select a feature variable with the greatest correlation to the FSV, add it to the selected feature variable subset (FS), and delete it from the FC.

Subsequently, a single feature variable was gradually selected from the FC set to join the FS subset. The SVR algorithm was used to establish the estimation model based on the FS subset, and the root mean square error (RMSE) of the estimation result of the model was calculated using leave-one-out cross-validation (LOOCV).

After traversing all candidate feature variables from the FC set, the feature variable subset with the smallest RMSE in this round was determined as the "optimal" feature variable subset, and the corresponding feature variables were added to the FS subset. The

iteration was continued until the specified number of iterations was reached or the RMSE of the FS subset was no longer reduced.

To compare with the proposed SVR-based feature variable selection method (FS-SVR), we used the SRA and FS-SVR methods to select feature variables for FSV estimation modeling. SRA is a feature selection method that constructs a multiple linear regression relationship between independent and dependent variables, gradually introducing new feature variables and eliminating feature variables that are not significant in the variance homogeneity test. In this study, the selection of feature variables based on the SRA method was completed using SPSS 23 software. The stepwise method was used to input the independent variable of SRA, and the significance level values of independent variable introduction and independent variable elimination were set to 0.05 and 0.1, respectively.

### 3.5. FSV Modeling and Accuracy Evaluation

The SVR algorithm was used to predict FSV values in the study area. In specific ap-plications, there remained some parameters to be adjusted, such as the kernel function and cost parameter (C). In the modeling process of the FSV estimation based on the SVR algorithm, the dependent variable was the observed value vector of the sample plot FSV, the independent variables were the selected remote sensing feature factor vectors, and the kernel functions used were the radial basis function (RBF) and linear kernel function (linear). Hyperparameter C indicates the tolerance of the estimation error. When using RBF, there is also a hyperparameter "gamma" that must be set. The larger the gamma, the fewer the support vectors. In the FSV estimation model based on SVR in this study, the value range of C was (1500), and the value range of gamma was (0.1, 5). By using cross-validation, the most suitable value for the training sample dataset was searched iteratively.

In this study, the LOOCV method was used to assess the accuracy of the predicted FSV and optimize the model [10,53]. In each iteration, one sample was selected as a test sample, and all remaining samples were used as a sample dataset for training the model parameters or hyperparameters. The LOOCV method can make full use of each FSV sample datum and can also greatly reduce the random error caused by the division of the training set and the validation set of the sample data. By comparing the predicted and observed values of the FSV, we obtained the coefficient of determination ($R^2$), RMSE, and relative root mean square error (rRMSE).

$$R^2 = 1 - \frac{\sum_{i=1}^{n}(y_i - \hat{y}_i)^2}{\sum_{i=1}^{n}(y_i - \overline{y})^2} \tag{9}$$

$$RMSE = \sqrt{\frac{\sum_{i=1}^{n}(y_i - \hat{y}_i)^2}{n}} \tag{10}$$

$$rRMSE = \frac{RMSE}{\overline{y}} \tag{11}$$

where $n$ is the sample size, $y_i$ is the measured FSV value of sample plot $i$, $\overline{y}$ is the mean value of the observed FSV of all sample plots, and $\hat{y}_i$ is the estimated FSV value of sample plot $i$. Generally, the higher the estimation accuracy of the model and the stronger the prediction ability, the greater the $R^2$ value and the smaller the RMSE and rRMSE values. Finally, the best feature variables and model were used to map the FSV of the coniferous plantations in the study area. In this study, The Sklearn machine learning model library were used to perform feature selection based on the FS-SVR method and the training and prediction of the FSV model.

## 4. Results

### 4.1. Feature Variables Extracted and Selected by Different Methods

In this study, optical remote sensing feature variables, such as the vegetation index and texture factors, were extracted based on the L8 OLI image. The 39 S1 images were preprocessed, and the backscatter coefficients were extracted. Three S1 images with similar

field investigation times and high correlations between the backscatter coefficient and FSV were selected: 20,190,711, 20,190,921, and 20,191,019. The corresponding texture feature factors were then extracted based on the backscattering coefficients of the three images. For the 32 pairs of S1 SLC im-ages that fulfilled the requirements of the spatial vertical baseline, a series of radar signal interference processing was conducted. Finally, 32 DSM and coherence coefficient images were generated. The InSAR coherence coefficient, and DSM images extracted from the Sentinel-1 SAR data are shown in Figure A1. For comparing and analyzing the estimation potential of the coniferous FSV of the L8, S1, and S1-InSAR remote sensing feature datasets, the feature variables were selected by SRA and FS-SVR based on the three datasets. As shown in Tables 5–8, for L8, S1, and S1-InSAR, the SRA method selects eight, six, and nine feature variables, respectively, and the FS-SVR method selects eight, eight, and ten feature factors, respectively.

**Table 5.** Statistical results using the SRA feature selection method based on the L8 OLI dataset.

| Variables | Coefficient | t | Sig. | 95% Confidence Interval | | Collinearity Statistics | | Correlation with FSV |
|---|---|---|---|---|---|---|---|---|
| | | | | Lower | Upper | Tolerance | VIF | |
| Constant | 66.627 | 1.654 | 0.101 | −13.129 | 146.383 | | | |
| b3-W5-T7 | 164.945 | 5.189 | 0.000 | 102.017 | 227.874 | 0.601 | 1.665 | 0.531 |
| b6-W9-T1 | −6.315 | −3.632 | 0.000 | −9.757 | −2.873 | 0.546 | 1.832 | −0.428 |
| RVI24 | −81.817 | −5.116 | 0.000 | −113.481 | −50.154 | 0.296 | 3.382 | −0.400 |
| RVI47 | 266.994 | 5.144 | 0.000 | 164.239 | 369.750 | 0.281 | 3.564 | −0.184 |
| b4-W5-T4 | 4.610 | 3.874 | 0.000 | 2.255 | 6.966 | 0.297 | 3.369 | −0.129 |
| b2-W7-T5 | −50.827 | −2.646 | 0.009 | −88.858 | −12.796 | 0.223 | 4.493 | −0.357 |
| b4-W5-T8 | 50.507 | 3.033 | 0.003 | 17.539 | 83.475 | 0.755 | 1.325 | −0.032 |
| b5-W9-T5 | 12.153 | 2.310 | 0.023 | 1.736 | 22.570 | 0.640 | 1.561 | −0.124 |

Note: b, band serial number of L8 image; W, size of texture analysis sliding window; T, serial number of texture factor (for example, b3-W5-T7 represents the seventh texture feature factor "Second Moment" extracted by band3 of the L8 image, and the sliding window size is 5 × 5); Correlation with FSV, the Pearson correlation between FSV observations and the feature variables.

**Table 6.** Statistical results using the SRA feature selection method based on the S1 SAR dataset.

| Variables | Coefficient | t | Sig. | 95% Confidence Interval | | Collinearity Statistics | | Correlation with FSV |
|---|---|---|---|---|---|---|---|---|
| | | | | Lower | Upper | Tolerance | VIF | |
| Constant | 108.650 | 4.181 | 0.000 | 57.212 | 160.087 | | | |
| 0921VH-B-W13-T1 | 196.682 | 7.254 | 0.000 | 143.009 | 250.354 | 0.070 | 14.193 | 0.142 |
| 0921VH-G-W9-T1 | −95.474 | −4.152 | 0.000 | −140.989 | −49.959 | 0.062 | 16.029 | 0.014 |
| 0711VV-S-W9-T1 | −49.330 | −4.976 | 0.000 | −68.956 | −29.705 | 0.159 | 6.290 | −0.095 |
| 0711VH-B-W15-T8 | 129.218 | 2.976 | 0.004 | 43.268 | 215.168 | 0.695 | 1.438 | 0.146 |
| 0828VH-S-W9-T8 | −102.517 | −3.402 | 0.001 | −162.157 | −42.876 | 0.697 | 1.435 | −0.091 |
| 0921VV-B-W13-T5 | 518.921 | 2.270 | 0.025 | 66.369 | 971.473 | 0.516 | 1.937 | −0.010 |

Note: S, Sigma$^0$; G, Gamma$^0$; B, Beta$^0$; W, size of texture analysis sliding window; T, serial number of the texture factor. For example, 0921VH-B-W13-T1 represents the first texture feature factor "mean" extracted by the backscattering coefficient, Beta$^0$, of the S1–20190921 VH polarization image, and the sliding window size was 13 × 13.

**Table 7.** Statistical results using the SRA feature selection method based on the S1-InSAR dataset.

| Variables | Coefficient | t | Sig. | 95% Confidence Interval | | Collinearity Statistics | | Correlation with FSV |
|---|---|---|---|---|---|---|---|---|
| | | | | Lower | Upper | Tolerance | VIF | |
| Constant | 447.447 | 5.463 | 0.000 | 285.288 | 609.606 | | | |
| 1206VV-CC | −200.281 | −3.052 | 0.003 | −330.227 | −70.335 | 0.570 | 1.755 | −0.600 |
| 1124VV-CC | −186.394 | −3.564 | 0.001 | −289.943 | −82.846 | 0.559 | 1.789 | −0.564 |
| 0711VV-DSM | 0.309 | 4.680 | 0.000 | 0.178 | 0.439 | 0.405 | 2.470 | 0.451 |
| 0703VV-CC | −72.551 | −2.813 | 0.006 | −123.617 | −21.485 | 0.947 | 1.056 | −0.061 |
| 0723VH-CC | −117.975 | −2.622 | 0.010 | −207.063 | −28.886 | 0.859 | 1.164 | −0.295 |
| 0621VV-CC | 94.824 | 2.640 | 0.009 | 23.710 | 165.939 | 0.835 | 1.197 | −0.122 |
| 1019VV-CC | −350.079 | −4.001 | 0.000 | −523.304 | −176.855 | 0.539 | 1.857 | −0.576 |
| 1108VV-CC | −159.089 | −2.811 | 0.006 | −271.156 | −47.021 | 0.498 | 2.008 | −0.567 |
| 1019VV-DSM | −0.147 | −2.287 | 0.024 | −0.275 | −0.020 | 0.293 | 3.408 | 0.491 |

Note: 1206, corresponding S1 image pair (20,191,206 and 20,191,230) in Table A2. For example, 1206VV-CC represents the coherence coefficient obtained by interference processing based on the S1 image pair "20,191,206 and 20,191,230" VV polarization.

**Table 8.** Statistical results using the FS-SVR feature selection method based on three datasets.

| Datasets | Variables | DC | PC | MIC | Importance |
|---|---|---|---|---|---|
| L8 OLI | b4-W9-T1 | 0.365 | −0.363 | 0.266 | 0.248 |
| | b1-W9-T7 | 0.363 | 0.294 | 0.356 | 0.148 |
| | b5-W3-T7 | 0.292 | 0.300 | 0.291 | 0.029 |
| | NDVI37 | 0.254 | −0.221 | 0.299 | 0.155 |
| | b6-W9-T7 | 0.365 | 0.338 | 0.298 | 0.148 |
| | b5-W9-T7 | 0.272 | 0.339 | 0.240 | 0.055 |
| | b4-W9-T7 | 0.388 | 0.392 | 0.295 | 0.095 |
| | b1-W7-T4 | 0.363 | −0.251 | 0.335 | 0.150 |
| S1 | 0711VV-B-W7-T1 | 0.184 | −0.146 | 0.272 | 0.370 |
| | 0921VH-B-W13-T1 | 0.183 | 0.142 | 0.239 | 0.488 |
| | 0711VH-G-W3-T7 | 0.164 | 0.067 | 0.213 | 0.067 |
| | 0711VH-B-W13-T4 | 0.163 | −0.031 | 0.296 | 0.115 |
| | 0921VH-S-W9-T3 | 0.160 | 0.109 | 0.230 | 0.071 |
| | 0711VV-B | 0.220 | −0.203 | 0.257 | 0.190 |
| | 0711VV-B-W13-T7 | 0.156 | 0.033 | 0.210 | 0.078 |
| | 0828VH-S-W9-T3 | 0.162 | 0.056 | 0.291 | 0.089 |
| S1-InSAR | 1206VV-CC | 0.590 | −0.600 | 0.415 | 0.194 |
| | 1124VV-CC | 0.544 | −0.564 | 0.401 | 0.126 |
| | 0617VV-CC | 0.280 | −0.059 | 0.301 | 0.034 |
| | 0703VV-CC | 0.261 | −0.061 | 0.358 | 0.075 |
| | 0921VV-CC | 0.273 | −0.080 | 0.310 | 0.034 |
| | 1108VV-CC | 0.581 | −0.567 | 0.439 | 0.130 |
| | 1019VV-CC | 0.581 | −0.576 | 0.508 | 0.160 |
| | 1214VV-CC | 0.361 | −0.340 | 0.314 | 0.034 |
| | 0317VH-CC | 0.269 | −0.132 | 0.264 | 0.042 |
| | 0422VH-DSM | 0.315 | −0.238 | 0.308 | 0.152 |

Note: DC, PC, and MIC are the distance correlation coefficient, Pearson correlation coefficient, and maximum information coefficient between feature variables and FSV observations, respectively. Importance: The importance of the feature variables measured by the random forest mean decreases in accuracy.

The collinearity statistics results (Tables 5–7) demonstrate that the feature variables of L8 and S1-InSAR have a smaller variance inflation factor (VIF) value and a higher tolerance value than the S1 backscatter and its texture factor as a whole. The texture features and vegetation index of L8, the coherence coefficient, and the DSM data of S1-InSAR had a good linear correlation with the FSV of the northern coniferous forest. Particularly notable was that the FSV correlation of 1206VV-CC and 1019VV-DSM reached 0.6 and 0.491, respectively, which were 311.0% and 236.3% higher than that of S1 (0711VH-b-W15-T8). In addition, for

the PC values between the FSV observations and the feature variables, the feature variables of L8 and S1-InSAR were significantly higher than those of S1.

*4.2. FSV Prediction Result*

Using the SVR linear kernel function and RBF kernel function algorithm, the coniferous forest stock volume estimation model of the Wangyedian Forest Farm was constructed based on the subset of feature variables selected by the SRA and FS-SVR methods (Figure 5). The prediction accuracies corresponding to the three datasets, L8, S1, and S1-InSAR, are listed in Table 9. In general, the S1-InSAR dataset had a better estimation performance for coniferous forest stock volume than that of S1 and L8. The combination of feature variables selected by the FS-SVR method is more suitable for coniferous FSV estimation than those selected by the SRA method. The average value of rRMSE obtained by FSV estimation based on the three datasets using the FS-SVR method was 23.17%, which was 13.8% lower than the average value of rRMSE corresponding to the SRA method (26.87%). When the SRA method was used, the FSV estimation performance of S1-InSAR and L8 was significantly higher than that of the S1 dataset, and the corresponding rRMSE values were 14.6% and 12.2% lower than that of S1, respectively, which also verified the correlation analysis results in Tables 5–7. For the FS-SVR method, the S1-InSAR feature factor set obtained by interferometry yielded good FSV estimation performance. The $R^2$, RMSE, and rRMSE values were 0.61, 47.2 $m^3$/ha, and 20.7%, respectively. To explore the synergistic potential of active and passive remote sensing combined data in northern coniferous forest stock volume estimation, we integrated the L8, S1, and S1-InSAR datasets and conducted FSV estimation experiments based on the FS-SVR method. The results show that the integrated dataset can provide full play to the synergy of the three remote sensing datasets (Figure 6). There was a strong correlation between the forest stock volume predicted values and observed values; the PC coefficient (r) value reached 0.81, and the RMSE and rRMSE values were 44.3 $m^3$/ha and 19.4%, respectively.

**Table 9.** Hyperparameter sets and prediction accuracies of support vector regression constructed by the SRA and FS-SVR methods.

| Datasets | Methods | C | Gamma | $R^2$ | RMSE ($m^3$/ha) | rRMSE (%) | r |
|---|---|---|---|---|---|---|---|
| L8 | SRA | 100 | | 0.38 | 59.0 | 25.9 | 0.63 |
| | FS-SVR | 150 | 2.8 | 0.40 | 58.4 | 25.6 | 0.63 |
| S1 | SRA | 200 | | 0.20 | 67.3 | 29.5 | 0.45 |
| | FS-SVR | 350 | 1.4 | 0.50 | 52.9 | 23.2 | 0.71 |
| S1-InSAR | SRA | 51 | | 0.42 | 57.3 | 25.2 | 0.65 |
| | FS-SVR | 150 | 4.8 | 0.61 | 47.2 | 20.7 | 0.78 |
| The integrated dataset | FS-SVR | 270 | 2.4 | 0.65 | 44.3 | 19.4 | 0.81 |

Note: The integrated dataset, the combination of L8, S1, and S1-InSAR datasets; r, Pearson correlation between FSV estimates and observations.

By using the FS-SVR method, the subset of feature variables selected based on the integrated dataset of L8, S1, and S1-InSAR and its estimation performance change trends are shown in Figure 7. The feature variables that contribute the most to the estimation accuracy of forest stock volume were InSAR coherence factors, followed by S1 backscattering texture factors and L8 OLI vegetation indices. With an increase in the number of feature variables, the estimation accuracy of the forest stock volume exhibits an obvious upward trend. Particularly notable is that, when the interference coherence factors 1202VV-CC and 0617VV-CC were added to the set of feature variables, the estimation accuracies (1-rRMSE) were 8.8% and 5.7% higher than those before the addition, respectively. The contribution of backscattering coefficient texture factor 0921VH-S-W9T3 to the improvement of forest stock volume estimation accuracy was relatively smaller than that of the feature factor related to VV polarization, which further verified the conclusion in Section 4.1; that is, the forest stock volume correlation of S1 VV polarization is higher than that of VH polarization.

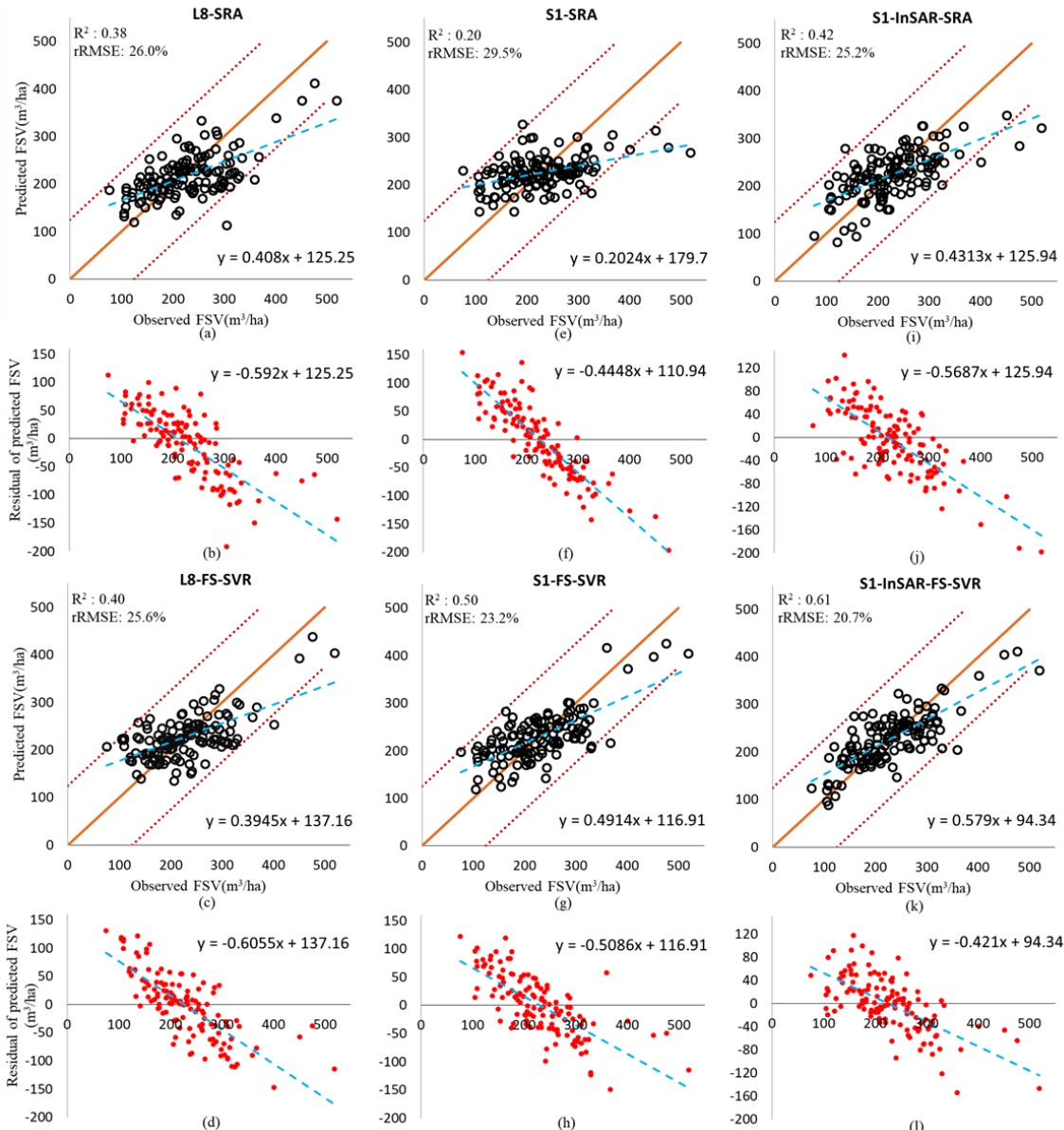

**Figure 5.** Scatterplots of FSV prediction results using L8 (**a**–**d**), S1 (**e**–**h**), and S1-InSAR (**i**–**l**) datasets, respectively. (**a**,**b**,**e**,**f**,**i**,**j**) SRA variable selection method. (**c**,**d**,**g**,**h**,**k**,**l**) FS-SVR variable selection method. The red dotted line is the 50% error bar, used to judge whether the error of the FSV estimated value exceeds 50% of the sample mean; the orange solid line is the 1:1 diagonal line; and the blue dotted line is the linear fitting trendline. The formula is the linear equation fitted by the scatterplot.

*4.3. Predicting and Mapping the Stock Volume of Coniferous Plantation*

To compare and analyze the stock volume estimation performance of different remote sensing datasets in the Wangyedian coniferous forest, we performed forest stock volume prediction and spatial distribution mapping based on the L8, S1, and S1-InSAR datasets (Figure 8a–c). The distribution range of the FSV estimates in Figure 8a was 112–336 $m^3$/ha, the low value of FSV was concentrated around 130 $m^3$/ha, and the high value was mainly concentrated around 300 $m^3$/ha, indicating that there were serious problems due to low-value overestimation and high-value underestimation, which also verified that the L8 image dataset suffers from a low saturation point in regions with high FSV values, resulting in very limited estimation capabilities. Figure 8b,c have larger FSV distribution ranges than Figure 8a, and their estimation results are also more accurate than those in Figure 8a. Especially in Figure 8c, the prediction effect in the low- and high-value

areas of the FSV was significantly improved, and it was more in line with the actual FSV distribution than Figure 8a,b. The aforementioned results show that the S1-InSAR dataset can effectively improve the estimation accuracy and saturation of the FSV in the Wangyedian coniferous forest.

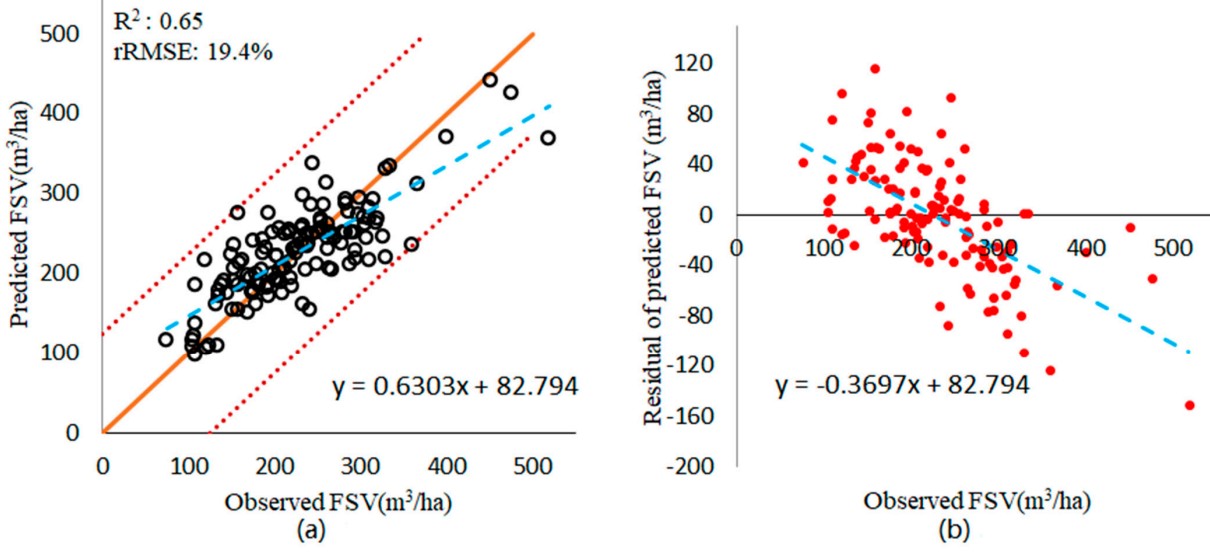

**Figure 6.** (**a**) Scatterplots of FSV prediction results based on integrated dataset of L8, S1, and S1-InSAR using the FS-SVR feature variable selection method and support vector regression model. (**b**) Residual distribution of FSV predicted value. The formula is the linear equation fitted by the scatterplot.

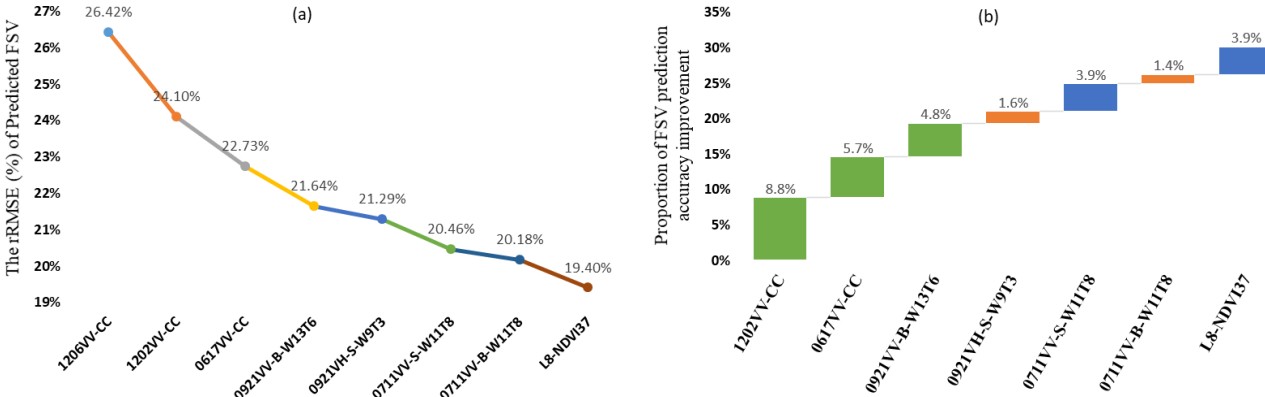

**Figure 7.** (**a**) FSV prediction rRMSE change trend with the increase in feature variables; (**b**) improvement trend of FSV prediction accuracy (1-rRMSE).

This study, using the integrated dataset of L8, S1, and S1-InSAR, combined with the sample data of the coniferous FSV in the study area, constructed an FSV estimation model using the FS-SVR method and an SVR algorithm. The coniferous FSV in the Wangyedian study area was predicted, and an FSV spatial distribution inversion map with 30 m resolution was generated (Figure 8d). The coniferous FSVs of the Wangyedian research area in 2019 ranged from 48 to 475 m³/ha, most of which were below 325 m³/ha, and the average value and standard deviation were 201 m³/ha and 62 m³/ha, respectively, which is consistent with the statistical results of our sampling survey. The low-value areas of coniferous FSV were concentrated in the central and northern low-altitude areas with many of human activities, and the high-value areas of the coniferous FSV were located in the relatively high-altitude or sparsely populated remote forest areas in the east and west, away from cities and towns.

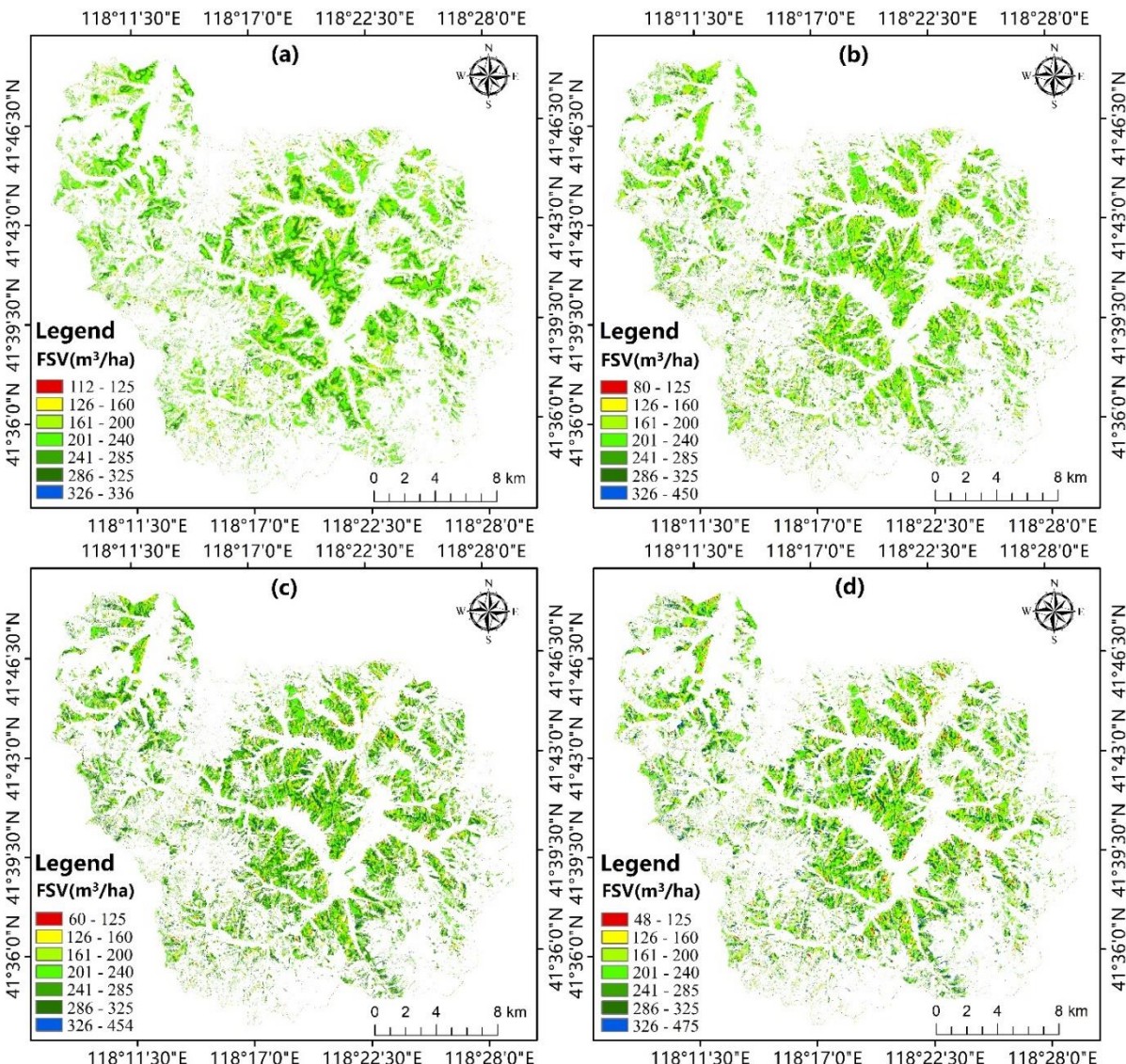

**Figure 8.** FSV distribution maps of coniferous plantations in the study area based on the FS-SVR method, using four remote sensing datasets. (**a**) L8, (**b**) S1, (**c**) S1-InSAR, (**d**) the integrated of three datasets.

## 5. Discussion

### 5.1. Feature Variables Selection in FSV Estimation

Unlike the SRA method, which focuses on selecting characteristic variables with good linear correlation, the FS-SVR method can measure the nonlinear relationship between feature variables, and FSV and considers the combination optimization effect between features. Therefore, the PC values of the feature variables in Table 8 were generally lower than those in Tables 5–7. However, the statistics of the FSV estimation results in Table 9 and Figure 5 demonstrate that the FS-SVR method proposed in this study has better FSV estimation accuracy than the SRA method does. Especially with respect to the FSV estimation based on the S1 dataset, the $R^2$ and fitting trendline slope corresponding to the FS-SVR method were 150.0% and 142.8% higher than those of the SRA method, respectively. These results also show that the combination effect between the feature variables is a crucial factor that cannot be ignored in the process of feature variable selection [12,50]. In addition, the correlation and importance statistics results in Tables 7 and 8 demonstrate that the VV polarization of the S1 image is significantly better than that of the VH polarization. In a study of forest AGB estimation based on optical and SAR remote sensing data in Dunhua City, Jilin Province, China, Chen [61] found that Sentinel-1 was more sensitive to forest

AGB under VV polarization data than HH polarization was. This finding is consistent with the results of this study.

*5.2. Analysis of FSV Estimation Performance*

We compared the estimation results of the FSV model in Figures 5 and 6 and found that the fitting effect of the model corresponding to the integrated dataset was the best, the slope of the fitting trendline (0.63) was the highest, and the fitting trendline of the FSV estimation residual value was the flattest. This finding shows that the synergy of the three remote sensing datasets can effectively suppress the problems of high-value underestimation and low-value overestimation in FSV estimation. The statistical results in Figure 7 demonstrate that the feature variable with the greatest improvement in the estimation accuracy of FSV in the three datasets is the S1-InSAR coherence coefficient, which may be because the coherent amplitude data contain the vertical structure information related to the forest canopy height. Therefore, the S1-InSAR dataset has a better FSV response and higher saturation point than the S1 backscattering coefficient and optical remote sensing factors do. Borlafmena et al. [45] tested the ability of the Sentinel-1 C-band image to distinguish forests from other land-use types and found that InSAR coherence feature factors can improve overall classification accuracy. Robert et al. [55] systematically evaluated the potential of Sentinel-1, Sentinel-2, and Landsat 8 for use in permanent grassland moving event detection. The results showed that comprehensive prediction accuracy based on the combined data of NDVI, the backscattering coefficient, and InSAR coherence was the highest. The findings of the aforementioned research are very similar to the conclusions of this study, which prove the great potential of the Sentinel-1 C-band and its InSAR coherence feature factors with respect to remote sensing classification and quantitative estimation.

Tomáš et al. [62] extracted radar polarization coherent amplitude data and optical remote sensing variables based on Sentinel-1 and Sentinel-2 data, supplemented by a sample survey and airborne laser scanning data, and established an AGB prediction model that used multiple regression. The best model RMSE was 41.2 t/ha, and the rRMSE was 35.1%. Long et al. [39] estimated the FSV of the Chinese fir plantation in South China by using Alos-2 PALSAR L-band full polarimetric SAR data. They found that the fused polarimetric features based on a timeseries of SAR images can improve the estimation accuracy of FSV, and the minimum rRMSE was 24.42%. In the aforementioned two studies, the difficulty of performing remote sensing prediction on forest AGB in reference [62] is essentially equivalent to that of the FSV estimation in this study. The Chinese fir forest studied in reference [39] is a coniferous plantation that contains larches and Chinese pines, similar to this study, but the FSV estimation accuracy (1-rRMSE) obtained in our study is 24.2% and 6.6% higher than those of the other two studies, respectively. The main reason for the above results is that we extracted many InSAR coherent amplitude factors based on hyper-temporal Sentinel-1 images, which improved the diversity of the remote sensing feature variables. In addition, we used the FS-SVR method to optimize the selection process of the feature variables.

*5.3. Limitations and Prospects*

There are many uncertain factors in the process of FSV remote sensing estimation that affect estimation accuracy, such as ground sample data, satellite sensor images, and feature variables for modeling [10,52]. The ground sample plots selected by sampling are representative. Too few sample plots cannot fully and accurately describe the actual distribution and law of the FSV [20]. Therefore, to minimize the errors caused by sampling, this study integrated field sample plot survey data from 2017 and 2019. There are some differences in the FSV saturation and estimation performance of different satellite sensor data. This study found that the estimated saturation point of L8 was approximately 300 $m^3$/ha, and the estimated maximum FSV value in the study area was 336 $m^3$/hal; whereas the saturation point of the S1-InSAR dataset was approximately 390 $m^3$/ha, the estimated maximum FSV value in the study area was 454 $m^3$/ha, and the best RMSE was

$47.6 \text{ m}^3/\text{ha}$. The aforementioned results are mainly because the optical sensor signal cannot penetrate the dense forest canopy, resulting in the spectral signal tending to be saturated [18, 20]. After interference processing of the S1 image results, the coherence coefficient and DSM data containing canopy height information can be obtained [42]. These S1-InSAR feature factors significantly improved the saturation and accuracy of FSV estimation. Because the interferometric coherence effect of the S1 images in the same period as the field survey is often not guaranteed, it is necessary to perform interferometric processing based on S1 hyper-temporal images to improve the diversity and effectiveness of coherent amplitude data. Such processing requires considerable time, and relying solely on InSAR technology cannot directly obtain tree height information in mountain forests [61]. ICESat-2 (Ice, Cloud, and Land Elevation Satellite-2) is a spaceborne LiDAR satellite that was launched in September 2018 [25]. Research [25,26] shows that ICESat-2 can obtain a large-scale forest vegetation canopy height. Therefore, for exploring effective solutions to further improve the accuracy of FSV remote sensing estimation, follow-up research can use ICESat2 data, combined with S1-InSAR and other optical remote sensing data, to synergistically retrieve the FSV. In addition, related research has used environmental factors, such as land surface temperature and soil moisture retrieved from Landsat 8 thermal infrared sensor images, to conduct remote sensing estimation and the modeling of parameters such as forest AGB and leaf area index [20,53,63]. However, the validity of environmental factors such as land surface temperature, soil moisture, and topographic moisture for estimating coniferous FSV has not been verified.

The set of remote sensing feature variables used to construct the FSV estimation model can significantly affect the final prediction results [10]. Pearson, SRA, and other linear relationship measurement methods can simply and quickly select feature variables linearly related to FSV. However, forests are characterized by spatial heterogeneity and dynamic changes. This method, based on linear correlation, cannot comprehensively and accurately describe the real relationship between the FSV and remote sensing variables in a complex forest environment [12]. In addition, the RF feature selection method can only screen the relatively crucial feature factors based on specific evaluation criteria, without considering the combination effect relationship between feature factors, which may be able to be coupled with other factors to some extent; thus, it has the potential to improve the accuracy of FSV estimation [50]. Compared with the SRA method, the FS-SVR method proposed in this paper has significant advantages in the FSV estimation of Wangyedian coniferous forests but needs to be verified in FSV research on other forest types in other research areas, and the operation efficiency of the algorithm needs to be improved. Nonparametric models represented by machine learning algorithms usually have better FSV estimation performance than parametric models do in a complex forest environment [10,50]. In this study, the estimation accuracy of the SVR model based on the RBF kernel function was higher than that based on the linear kernel function. However, due to the spatial heterogeneity of forest ecosystems, there remains a certain estimation error in the FSV estimated by the SVR model (rRMSE, 19.4%). The hybrid model, combining the parametric or nonparametric model and the residual Kriging interpolation model, showed excellent performance in forest AGB estimation [61], and the RMSE reached 29.72 Mg/ha. In further research, a Kriging interpolation model based on geostatistics can be considered for FSV prediction research.

## 6. Conclusions

In this study, L8 OLI multispectral images, C-band S1, and interferometric radar data were used to extract various remote sensing feature factors, and an improved method for the selection of remote sensing feature variables (FS-SVR) was explored. On the basis of three datasets and the SVR model, a FSV remote sensing estimation experiment on coniferous forest was conducted, and an FSV estimation scheme, combined with active and passive multisource remote sensing data, was proposed. Finally, spatial distribution inversion mapping of the coniferous FSV in the Wangyedian Forest Farm was conducted.

The conclusions of this study are as follows: (1) The S1-InSAR dataset generated based on S1 hyper-temporal image interferometry had good FSV estimation accuracy. The $R^2$ reached 0.6, the RMSE value was 47.6 m$^3$/ha, and the rRMSE value was 20.9%. Because the coherent amplitude and DSM data contain the vertical structure information related to the forest canopy height, and the hyper-temporal S1 image data greatly enriches the diversity of S1-InSAR feature factors, they have a better FSV response and a higher saturation point than remote sensing factors do, such as the S1 backscattering coefficient and the L8 vegetation index. (2) The integrated dataset of L8, S1, and S1-InSAR can fully play the synergy of the three remote sensing datasets. The RMSE and rRMSE values are 44.3 m$^3$/ha and 19.4%, respectively. There is a strong correlation between the FSV predicted value and observed value, and r reaches 0.81, which is 28.6%, 14.1%, and 3.8% higher than those of L8, S1, and S1-InSAR, respectively. The feature variables that contribute the most to the accuracy of FSV estimation in the integrated dataset are InSAR coherence factors, followed by S1 backscattering coefficient texture factors and L8 vegetation indices. The backscattering coefficient and the InSAR feature factor of the VV polarization in the S1 image provide better FSV estimation performance than the VH polarization. (3) The proposed FS-SVR method is very suitable for the selection of remote sensing features in FSV estimation. The average value of rRMSE (23.17%) obtained using the FS-SVR method for FSV estimation based on the three datasets was 13.8% lower than that of the SRA method (26.87%). This study helps realize the high-precision estimation and mapping of regional forest volumes by collecting a small amount of measured forest plot data combined with global coverage, free-download multispectral images (Landsat 8), and C-band SAR data (Sentinel-1), which have practical theoretical guidance and practical demonstration significance for the development of remote sensing estimation technology for regional forest volumes, and will promote the further improvement of modern forestry resources management.

**Author Contributions:** Conceptualization, X.L. and Z.Y.; methodology, X.L., J.L., Z.Y. and H.L.; software, X.L.; validation, X.L., H.Z. and Z.Y.; formal analysis, X.L. and H.L.; investigation, X.L., J.L., Z.Y. and H.L.; resources, X.L., J.L. and Z.Y.; data processing, X.L., J.L., Z.Y. and H.Z.; original draft, X.L.; review and revision, X.L., H.L. and Z.Y.; final editing: X.L.; visualization, X.L. and Z.Y.; supervision, H.L.; project administration, X.L.; funding acquisition, X.L., H.L. and J.L. All authors have read and agreed to the published version of the manuscript.

**Funding:** This study was financially supported by the National Natural Science Foundation of China (N#:32171784); the Excellent Youth Project of the Scientific Research Foundation of the Hunan Provincial Department of Education (N#: 21B0808); the Hunan Provincial Natural Science Foundation of China (N#: 2021JJ31158); the Changsha Natural Science Foundation (N#: kq2202315); and the National Key R&D Program of China project "Research of Key Technologies for Monitoring Forest Plantation Resources" (N#:2017YFD0600900).

**Data Availability Statement:** The observed GSV data from the sample plots and spatial distribution data of forest resources presented in this study are available on request from the corresponding author. Those data are not publicly available due to privacy and confidentiality. The Landsat 8 and Sentinel-1 images were obtained from the United States Geological Survey (https://earthexplorer.usgs.gov/) (accessed on 9 May 2021) and the European Space Agency Copernicus data center (https://scihub.copernicus.eu/) (accessed on 11 July 2021), respectively. DEM data covering the study area were downloaded from the geospatial data cloud (http://www.gscloud.cn/) (accessed on 8 May 2019).

**Conflicts of Interest:** The authors declare no conflict of interest.

## Appendix A

**Table A1.** Landsat 8 and Sentinel-1 image information covering the Wangyedian research area used in this study.

| Image Category | Image Identification | Acquisition Date |
|---|---|---|
| Landsat 8 | LC08_L1TP_122031_20190927_20191017_01_T1 | 20190927 |
| | S1B_IW_SLC__1SDV_20191230T221132_20191230T221159_019600_0250B0_2899 | 20191230 |
| | S1B_IW_SLC__1SDV_20191218T221133_20191218T221200_019425_024B1B_4D83 | 20191218 |
| | S1B_IW_SLC__1SDV_20191206T221133_20191206T221200_019250_024587_75B0 | 20191206 |
| | S1B_IW_SLC__1SDV_20191124T221134_20191124T221201_019075_023FFC_F9D2 | 20191124 |
| | S1B_IW_SLC__1SDV_20191112T221134_20191112T221201_018900_023A5C_5D48 | 20191112 |
| | S1B_IW_SLC__1SDV_20191019T221134_20191019T221201_018550_022F3E_5B86 | 20191019 |
| | S1B_IW_SLC__1SDV_20191007T221134_20191007T221201_018375_0229DE_E289 | 20191007 |
| | S1B_IW_SLC__1SDV_20190925T221134_20190925T221201_018200_022459_AE9B | 20190925 |
| | S1B_IW_SLC__1SDV_20190901T221133_20190901T221200_017850_02197D_522E | 20190901 |
| | S1B_IW_SLC__1SDV_20190820T221132_20190820T221159_017675_02140F_526D | 20190820 |
| | S1B_IW_SLC__1SDV_20190810T095659_20190810T095725_017522_020F3E_BC60 | 20190810 |
| Sentinel-1B | S1B_IW_SLC__1SDV_20190808T221500_20190808T221159_017500_020E97_6E25 | 20190808 |
| | S1B_IW_SLC__1SDV_20190727T221131_20190727T221158_017325_02094E_08A7 | 20190727 |
| | S1B_IW_SLC__1SDV_20190715T221130_20190715T221157_017150_020438_6966 | 20190715 |
| | S1B_IW_SLC__1SDV_20190703T221129_20190703T221156_016975_01FF10_9955 | 20190703 |
| | S1B_IW_SLC__1SDV_20190621T221129_20190621T221156_016800_01F9E2_C83C | 20190621 |
| | S1B_IW_SLC__1SDV_20190609T221128_20190609T221155_016625_01F4AC_6072 | 20190609 |
| | S1B_IW_SLC__1SDV_20190528T221127_20190528T221154_016450_01EF78_A431 | 20190528 |
| | S1B_IW_SLC__1SDV_20190516T221127_20190516T221154_016275_01EA17_1542 | 20190516 |
| | S1B_IW_SLC__1SDV_20190504T221126_20190504T221153_016100_01E499_637A | 20190504 |
| | S1B_IW_SLC__1SDV_20190422T221126_20190422T221153_015925_01DEBD_37CA | 20190422 |
| | S1B_IW_SLC__1SDV_20190329T221125_20190329T221152_015575_01D321_91C4 | 20190329 |
| | S1B_IW_SLC__1SDV_20190317T221124_20190317T221151_015400_01CD68_5402 | 20190317 |
| | S1A_IW_SLC__1SDV_20191226T095744_20191226T095812_030518_037E99_1211 | 20191226 |
| | S1A_IW_SLC__1SDV_20191214T095744_20191214T095812_030343_037891_1BF4 | 20191214 |
| | S1A_IW_SLC__1SDV_20191202T095745_20191202T095813_030168_037286_5B2A | 20191202 |
| | S1A_IW_SLC__1SDV_20191120T095745_20191120T095813_029993_036C78_D64C | 20191120 |
| | S1A_IW_SLC__1SDV_20191108T095745_20191108T095813_029818_036669_AAD4 | 20191108 |
| | S1A_IW_SLC__1SDV_20191015T095745_20191015T095813_029468_035A34_B673 | 20191015 |
| | S1A_IW_SLC__1SDV_20190921T095745_20190921T095813_029118_034E23_9420 | 20190921 |
| | S1A_IW_SLC__1SDV_20190909T095745_20190909T095812_028943_034826_7BA8 | 20190909 |
| Sentinel-1A | S1A_IW_SLC__1SDV_20190828T095744_20190828T095812_028768_034210_B0B1 | 20190828 |
| | S1A_IW_SLC__1SDV_20190816T095743_20190816T095811_028593_033BF0_21BC | 20190816 |
| | S1A_IW_SLC__1SDV_20190804T095743_20190804T095811_028418_033619_59BC | 20190804 |
| | S1A_IW_SLC__1SDV_20190723T095742_20190723T095810_028243_0330C2_ECFE | 20190723 |
| | S1A_IW_SLC__1SDV_20190711T095741_20190711T095809_028068_032B79_A792 | 20190711 |
| | S1A_IW_SLC__1SDV_20190629T095740_20190629T095808_027893_03262C_B424 | 20190629 |
| | S1A_IW_SLC__1SDV_20190617T095739_20190617T095807_027718_0320F1_5DB1 | 20190617 |
| | S1A_IW_SLC__1SDV_20190605T095739_20190605T095807_027543_031BAA_448C | 20190605 |

**Table A2.** Pairing of 39 Sentinel-1 IW SLC images covering Wangyedian study area in 2019.

| Number | Paired Image | Satellite and Track | Vertical Baseline Length (m) | Time Baseline Interval (Days) | Interference Processing |
|---|---|---|---|---|---|
| 1 | 20190317 and 20190329 | S1B-Descending | 12.462 | 12 | No |
| 2 | 20190317 and 20190422 | S1B-Descending | 46.479 | 36 | Yes |
| 3 | 20190329 and 20190422 | S1B-Descending | 58.387 | 24 | Yes |
| 4 | 20190422 and 20190504 | S1B-Descending | 81.720 | 12 | Yes |
| 5 | 20190422 and 20190516 | S1B-Descending | 35.526 | 24 | Yes |
| 6 | 20190504 and 20190516 | S1B-Descending | 48.376 | 12 | Yes |
| 7 | 20190504 and 20190528 | S1B-Descending | 74.800 | 24 | Yes |
| 8 | 20190516 and 20190528 | S1B-Descending | 26.943 | 12 | Yes |
| 9 | 20190605 and 20190617 | S1A-Ascending | 46.979 | 12 | Yes |
| 10 | 20190609 and 20190621 | S1B-Descending | 55.835 | 12 | Yes |
| 11 | 20190617 and 20190629 | S1A-Ascending | 74.956 | 12 | Yes |
| 12 | 20190621 and 20190703 | S1B-Descending | 91.912 | 12 | Yes |
| 13 | 20190629 and 20190711 | S1A-Ascending | 11.979 | 12 | No |
| 14 | 20190703 and 20190715 | S1B-Descending | 31.498 | 12 | Yes |

**Table A2.** *Cont.*

| Number | Paired Image | Satellite and Track | Vertical Baseline Length (m) | Time Baseline Interval (Days) | Interference Processing |
|---|---|---|---|---|---|
| 15 | 20190711 and 20190723 | S1A-Ascending | 58.018 | 12 | Yes |
| 16 | 20190715 and 20190727 | S1B-Descending | 69.089 | 12 | Yes |
| 17 | 20190723 and 20190804 | S1A-Ascending | 70.665 | 12 | Yes |
| 18 | 20190727 and 20190808 | S1B-Descending | 3.070 | 12 | No |
| 19 | 20190804 and 20190816 | S1A-Ascending | 12.535 | 12 | No |
| 20 | 20190808 and 20190820 | S1B-Descending | 34.460 | 12 | Yes |
| 21 | 20190816 and 20190828 | S1A-Ascending | 12.676 | 12 | No |
| 22 | 20190820 and 20190901 | S1B-Descending | 10.237 | 12 | No |
| 23 | 20190828 and 20190909 | S1A-Ascending | 65.810 | 12 | Yes |
| 24 | 20190901 and 20190925 | S1B-Descending | 42.950 | 24 | Yes |
| 25 | 20190909 and 20190921 | S1A-Ascending | 17.108 | 12 | Yes |
| 26 | 20190921 and 20191015 | S1A-Ascending | 149.681 | 24 | Yes |
| 27 | 20190925 and 20191007 | S1B-Descending | 85.305 | 12 | Yes |
| 28 | 20191007 and 20191019 | S1B-Descending | 46.193 | 12 | Yes |
| 29 | 20191015 and 20191108 | S1A-Ascending | 111.245 | 24 | Yes |
| 30 | 20191019 and 20191112 | S1B-Descending | 68.399 | 24 | Yes |
| 31 | 20191108 and 20191120 | S1A-Ascending | 30.156 | 12 | Yes |
| 32 | 20191112 and 20191124 | S1B-Descending | 8.064 | 12 | No |
| 33 | 20191112 and 20191206 | S1B-Descending | 82.270 | 24 | Yes |
| 34 | 20191120 and 20191202 | S1A-Ascending | 57.503 | 12 | Yes |
| 35 | 20191124 and 20191206 | S1B-Descending | 74.109 | 12 | Yes |
| 36 | 20191202 and 20191214 | S1A-Ascending | 49.938 | 12 | Yes |
| 37 | 20191206 and 20191230 | S1B-Descending | 60.980 | 24 | Yes |
| 38 | 20191214 and 20191226 | S1A-Ascending | 44.198 | 12 | Yes |
| 39 | 20191218 and 20191230 | S1B-Descending | 55.525 | 12 | Yes |

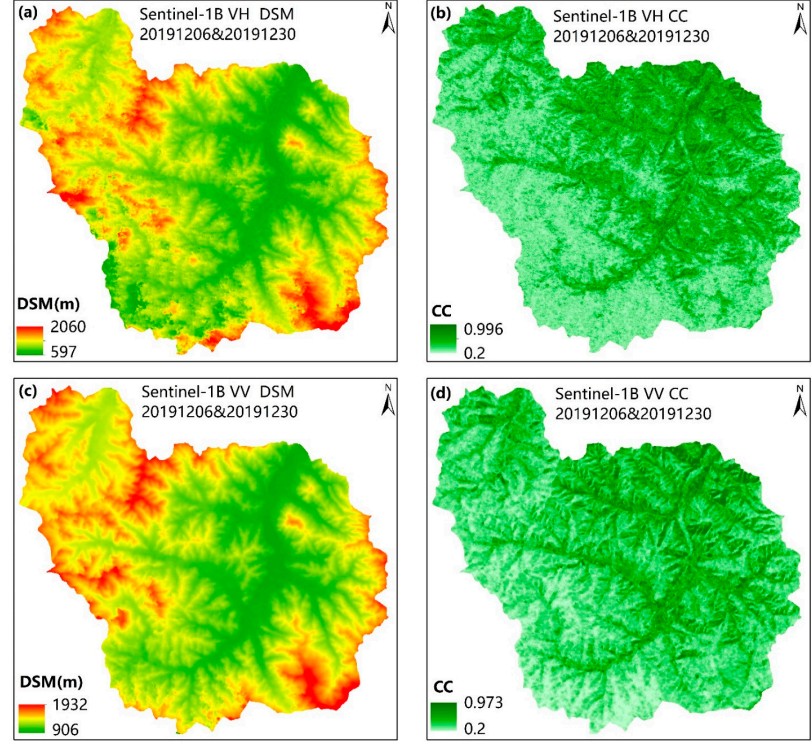

**Figure A1.** DSM and CC images generated by InSAR processing based on Sentinel-1 image pair (20191206 and 20191230). (**a**,**c**) DSM generated based on VH and VV polarized images, respectively; (**b**,**d**) CC maps generated based on VH and VV polarized images, respectively.

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
