# Peer review of "Inversion of Coniferous Forest Stock Volume Based on Backscatter and InSAR Coherence Factors of Sentinel-1 Hyper-Temporal Images and Spectral Variables of Landsat 8 OLI"

_remotesensing, doi:10.3390/rs14122754_

Round 1

Reviewer 1 Report

The reviewed work concerns the application of various remote methods in assessing forest stock volume. Like most works in this field, it is general and does not always reflect the reality of the biological productivity of forest ecosystems (not only). It has significance for forest management (forest stock, range, and degree of forest health). Comments below for correction and consideration: 

Abstract:

The abstract is very long and confusing. There is a lack of work goals here. It should be shortened by 20%. 

Keyword - should not be the same as in the title. Please change them. 

Introduction - the introduction is also long, and the focus should be on the research topic. The section on LIDAR should be removed (lines 77-84). The paper's objectives should be stated at the end of the introduction there are none. Give the purposes of the work at the end of the introduction, as they are not there now. 

Study area:
150-151 m3 line (give as superscript)
Give more information about forests (natural and anthropogenic) - species composition as this is the main object of this work. Species names should be given in Latin names. 
The quality of Fig 1 is poor. 
Sample plot design and FSV data collection:
I am not sure that the plot size for such an area is representative. 

The other methods of obtaining data are not questionable. 
Methods:
Line 222 - green plant change to plant.

Lines 218-234: the description is not about the research method but rather about the discussion. I propose to delete this section (preferably) or move it to the discussion. 

Results: presented relatively clearly, the usefulness of the methods used was discussed.  The theme of the forests studied disappeared somewhere. 

The conclusions lack information on the methods used in forest management and their applicability.  It would be an essential conclusion.

Reviewer 2 Report

The article “Inversion of Coniferous forest stock volume based on backscatter and InSAR coherence factors of Sentinel-1 hyper-temporal images and spectral variables of Landsat 8 OLI”, deals with the coniferous forest in China and its implication in the assimilation of CO2, for which the biomass is studied through sensors.

The study seems interesting and the technique used is correct, despite missing the characteristics of the forests studied, their structure, diversity, dominant species; I have not seen if this type of coniferous forest studied is natural or repopulated, this should be clarified.

In order to make reading more comfortable and legible for potential readers, we recommend that the meaning of some formulas be more clearly expressed.

Reviewer 3 Report

On a general note, the paper is quite interesting and can contribute to the growing literature using remote sensing data estimating forest biomass and the carbon sink. Although, the authors need to revise their work and address the following comments:

The clarity of drawings and graphs are not very good.

Can you give us more details about remote sensing estimation methods and especially about the Figure 2 c and Figure 3. Flow chart of interference processing based on Sentinel-1 IW SLC images?

Conclusions could be the base and a direction for future research. What are your future proposals?

In page 2  line 51 the word ecosystem needs an s to be correct.

In the page 3 line 137 the words"not easy to ovefit"can be better expressed with the words "resilient to overfiting" 

In page 4 line 157 the word "determined "is better to changed with the word "selected"

In page 8 line 263, the sentence begings with small letter w,onstead of capital.

In page 10 lines356-358 the hole sentence needs improving 
